# Sprouted Barley Flour as a Nutritious and Functional Ingredient

**DOI:** 10.3390/foods9030296

**Published:** 2020-03-05

**Authors:** Daniel Rico, Elena Peñas, María del Carmen García, Cristina Martínez-Villaluenga, Dilip K. Rai, Rares I. Birsan, Juana Frias, Ana B. Martín-Diana

**Affiliations:** 1Subdirection of Research and Technology, Agro-Technological Institute of Castilla y León, Consejería de Agricultura y Ganadería, Finca de Zamadueñas, 47171 Valladolid, Spain; ricbarda@itacyl.es (D.R.); gargurmar@itacyl.es (M.d.C.G.); mardiaan@itacyl.es (A.B.M.-D.); 2Department of Food Characterization, Quality and Safety, Institute of Food Science, Technology and Nutrition (ICTAN-CSIC), 28006 Madrid, Spain; c.m.villaluenga@csic.es (C.M.-V.); frias@ictan.csic.es (J.F.); 3Department of Food BioSciences, Teagasc Food Research Centre Ashtown, 15 Dublin, Ireland; Dilip.Rai@teagasc.ie (D.K.R.); Rares.Birsan@teagasc.ie (R.I.B.)

**Keywords:** barley, germination, flour, RSM, nutritional properties, bioactive compounds, quality

## Abstract

The increasing demand for healthy food products has promoted the use of germinated seeds to produce functional flours. In this study, germination conditions were optimized in barley grains with the aim to produce flours with high nutritional and biofunctional potential using response surface methodology (RSM). The impact of germination time (0.8–6 days) and temperature (12–20 °C) on barley quality was studied. Non-germinated barley was used as the control. The content of vitamins B_1_, B_2_ and C, and proteins increased notably after germination, especially at longer times, while levels of fat, carbohydrates, fibre, and β-glucan were reduced. Total phenolic compounds, γ-aminobutyric acid and antioxidant activity determined by Oxygen Radical Absorbance Capacity increased between 2-fold and 4-fold during sprouting, depending on germination conditions and this increase was more pronounced at higher temperatures (16–20 °C) and longer times (5–6 days). Procyanidin B and ferulic acid were the main phenolics in the soluble and insoluble fraction, respectively. Procyanidin B levels decreased while bound ferulic acid content increased during germination. Germinated barley flours exhibited lower brightness and a higher glycemic index than the control ones. This study shows that germination at 16 °C for 3.5 days was the optimum process to obtain nutritious and functional barley flours. Under these conditions, sprouts retained 87% of the initial β-glucan content, and exhibited levels of ascorbic acid, riboflavin, phenolic compounds and GABA between 1.4-fold and 2.5-fold higher than the non-sprouted grain.

## 1. Introduction

Nowadays, the main changes in demand for agricultural and food products are being fueled by population growth and lifestyle modifications. Many food manufacturers today are looking to replace wheat flour by alternative flours to be included in new formulations -with high nutritive value and bioactive properties.

Barley *(Hordeum vulgare* L.) is an ancient cereal which has been traditionally used for animal feeding and as raw material in malting industry [1]. Today, barley remains an important grain in some cultures of Asia and Northern Africa, and in recent years, there has been an increasing interest in using barley as an ingredient in food products due its nutritional value and high content of biologically active compounds [2]. This renewed trend in formulating novel barley foods is based on the beneficial effects of β-glucans on reducing blood cholesterol and glycemic index. In fact, the European Commission has authorized health claims linking barley β-glucan with a reduction of blood cholesterol. This claim can be used in foods with β-glucan levels ≥1 g [3]. The high β-glucan content in barley grains makes it an attractive ingredient for development of novel functional flours [4]. In addition to β-glucan, barley contains some other bioactive compounds (phenolic compounds, tocols and sterols) with beneficial effects that are currently under extensive investigation [5]. Therefore, there is a large potential to include barley flour partially or as a whole grain in a wide range of cereal-based food products as an alternative to commonly used cereals (wheat, rice, maize and oat).

Most barley varieties are hulled, although several hull-less varieties are also cultivated. Hull-less barley, known as ‘naked’ barley, requires minimal cleaning compared to hulled barley, as no processing is needed to remove the inedible outer hull [6]. The use of hull-less barley varieties with high levels of β-glucans is interesting to obtain functional barley flours that can be easily incorporated into foods to meet the authorized health claims of β-glucan. Barley flours usually have a higher water absorption ability than wheat flours for its greater content of soluble fiber [7], which can positively affect their baking properties.

Nowadays, there is a strong demand for food products with health-promoting properties. This is reflected by the growing market of foods that contain functional ingredients able to modulate different body functions, improving the health of consumers [8]. The scientific community is focusing its efforts on identifying novel functional ingredients that will expand the options to develop innovative food products with health benefits. Barley flours enriched in bioactive compounds are attractive ingredients to achieve these purposes, since they can be easily incorporated into different food matrices. Germination can be explored as a low-cost process to enhance the levels of functional compounds with healthy attributes in barley. Germination has been demonstrated to be an inexpensive and sustainable process that improves the nutritive quality and the content of functional compounds of grains, but also their palatability, digestibility and bioavailability [9,10]. However, the magnitude of changes caused by germination depends on the grain variety and germination conditions. For this reason, the optimization of germination parameters is crucial for improving the nutritional and bioactive properties of a selected grain variety. Although different studies have investigated the influence of germination in several cereals and legumes, few studies have optimized germination conditions in barley for producing high-quality flours. Thus, the objective of this study was to explore the effect of germination temperature and time on the nutritional, bioactive and other quality features in a hull-less barley variety. The functional sprouted flour obtained could be used for the production of functional bakery products (bread, cookies, cakes, muffins, etc).

## 2. Materials and Methods

### 2.1. Chemicals

α-Amylase from porcine pancreas (EC 3.2.1.1), 2,2’-Azinobis 3-ethylbenzothiazoline-6-sulfonic acid (ABTS), 2,2’-diazobis-(2-aminodinopropane)-dihydrochloride (AAPH), 2,2-diphenyl-1-picrylhydrazyl (DPPH), Folin–Ciocalteu (FC) reagent, gallic acid (GA), 6-hydroxy-2,5,7,8-tetramethyl-2-carboxylic acid (Trolox), and sodium carboxymethylcellulose (CMC) were obtained from Sigma-Aldrich, Co. (St. Louis, MO, USA). Amyloglucosidase (EC 3.2.1.3), glucose oxidase-peroxidase (GOPOD) and 1.3:1.4 mixed-linkage β-glucan kits were provided by Megazyme International Ireland (Wicklow, Ireland).

### 2.2. Barley Grains

A hull-less H13 barley variety obtained by conventional breeding between Merlin and Volga varieties in Agro-Technological Institute of Castilla y León was used. This barley variety, commercially known as GALIS, contains high levels of β-glucan (5.2 g/100 g). Whole-grain flour was obtained in order to compare the results obtained in sprouted grains with those of raw grain andevaluate the changes in nutrients, bioactive compounds and other quality parameters caused by the germination process. For these purposes, whole grains were milled by using a mechanical grinder (Moulinex, France), sifted through a 60-mesh sieve (0.5 mm pore diameter) and stored under vacuum at −20 °C. This flour was also analyzed.

### 2.3. Germination

Twenty grams of barley seeds was soaked in 0.1% sodium hypochlorite at 1:5 ratio (*w*/*v*) for 30 min and then washed with tap water until it reached a neutral pH. Then, seeds were steeped in tap water (at the same ratio) at room temperature for 4 h. Water was removed and seeds were spread across moist filter over steel grid, which was placed in plastic trays containing sterile tap water. Seeds were covered by a filter paper and introduced in a germination cabinet (model MLR-350, Sanyo, Japan), with >90% air humidity. Germination was carried out in darkness at temperatures in the range 12.1–19.9 °C and times of 1.6–6.19 days, conditions chosen according to the response surface model (Table 1). The germination process was carried out in triplicate for each germination condition. The sprouted seeds were freeze-dried (LyoQuest, Telstar, Spain), milled (Moulinex, France), sifted through a 60-mesh sieve (0.5 mm pore diameter) and stored under vacuum at −20 °C until further analysis.

### 2.4. Experimental Design

The response surface methodology (RSM) was employed to evaluate the impact of germination time and temperature on different quality and bioactive properties of sprouted barley. A central composite design was used with a total of 11 experimental assays, including three center points and “star” points to estimate curvature. The order of the experiments was randomly selected to avoid the effects of lurking variables (Table 1 and Figure 1). The combination of the two factors (germination temperature and time) studied in the response surface design and its optimization was based on the +1 and −1 variable levels. Proximal, nutritional, bioactive and sensory parameters were selected as dependent variables. The design was used to explore the quadratic response surfaces and to obtain polynomial equations for each response variable. The optimization of germination conditions was performed through the desirability function approach that allows identifying the most suitable combination of germination time and temperature to achieve the maximum desirable response values.

### 2.5. Proximate and Fatty Acid Compositions

The moisture contents of raw and germinated flours obtained from barley were determined by drying 5 ± 0.001 g sample flour at 105 °C for 3 h. Total protein content was assessed by the Dumas method. A factor of 6.25 was used to transform nitrogen into protein values. Total fat content was evaluated after extraction of sample flour with petroleum ether (BP 40–60 °C) during 4 h in a Soxtec fat-extracting unit. Carbohydrates were estimated by difference. Total dietary fiber content was determined by a commercial kit provided (Sigma, St. Louis, MO, USA), by following the manufacturer’s instructions. Ash content was evaluated by incineration of samples in a muffle furnace at 550 °C for 5 h. The results were expressed as g/100 g of dry matter (d.m.), after correcting them for moisture content.

For the evaluation of fatty acid composition in sample flours, the lipid-containing chloroform phase was separated and evaporated. Then, the sample was dissolved in 1 mL of hexane and methylated by the addition of 0.5 M methanolic KOH (100 μL) and incubation for 10 min at room temperature. An analysis of fatty acid methyl esters (FAME) in the upper layer obtained was carried out on a gas chromatograph (Agilent 7890A, Agilent Technologies, CA, USA) coupled with a flame ionization detector.

### 2.6. Content of Vitamin C

Vitamin C content was determined in barley flour samples by homogenization of the sample with 50 mL of oxalic acid (2 g/100 g) and titration with 2,6,-dichlorolindophenol (0,25 g/L) using a potentiometric titrator (877 Tritino Plus, Metrohm, Switzerland). The results were expressed as mg vitamin/100 g d.m.

### 2.7. Content of Thiamine and Riboflavin

Both vitamins were extracted from raw and germinated barley flour by acid and enzymatic hydrolysis and quantified by reversed-phase high-performance liquid chromatography (HPLC) coupled to a fluorescence detector, according to Frias et al. [11]. The results were expressed as μg vitamin/100 g d.m.

### 2.8. Content of β-Glucan

The β-glucan content was quantified by 1.3:1.4 mixed-linkage β-glucan kit (Megazyme, Ireland), following the manufacturer instructions. The assay uses lichenase and β-glucosidase to hydrolyse β-glucan to glucose. Subsequently, glucose reacts with GOPOD (glucoseoxidase/peroxidase) reagent and the absorbance at 510 nm of the resultant compound was measured in a microplate reader Synergy HT (BioTek Instruments, Winooski, VT, USA). All measurements were performed in triplicate. The results were expressed as g β-glucan/100 g d.m.

### 2.9. Content of γ-Aminobutyric Acid (GABA)

GABA was quantified in water extracts obtained from raw and germinated barley samples using HPLC coupled to a diode array detector as previously described [9]. Results were expressed as mg/100 g d.m.

### 2.10. Total Phenolic Content Determined by the Folin–Ciocalteu Method

Total phenolic contents (TPCs) were evaluated in methanolic extracts obtained from control and sprouted barley flours. Briefly, 1 g of sample was extracted with 10 mL of methanol:water (1:1, *v*/*v*; acidified to pH 2 with 0.1 M HCl) by stirring (250 rpm, 25 °C, 1 h) in an orbital shaker. Samples were centrifuged (25 °C, 3800× *g*, 10 min), and the supernatant was collected, filtered (Whatman paper n° 1), and made into a final volume of 25 mL with the extracting solvent. Extracts were stored at −80 °C until use. TPCs were estimated using the Folin–Ciocalteu method as previously reported [12]. Gallic acid (GA) was used as the standard. The final results were corrected for moisture and expressed as µmol GA equivalents (GA Eq)/g d.m.

### 2.11. Content and Profile of Free and Bound Phenolic Compounds

Free phenolic compounds were extracted as reported above (see Section 2.9). The residues obtained following the free phenolic extraction were subjected to alkaline and acid hydrolysis for recovering the bound (insoluble) fraction, as described by Mattila et al. [13]. Soluble and insoluble phenolic extracts were filtered through a 0.22-µm filter, lyophilized and stored at −20 °C until analysis.

The profile of free and bound phenolic compounds in barley samples was analyzed by HPLC coupled with a mass spectrometer, according to Birsan et al. [14]. The results were expressed as µg/mg of dry extract.

### 2.12. Total Antioxidant Activity

Total antioxidant activity was measured in methanolic extracts previously obtained using classical assays, namely DPPH radical scavenging activity, Oxygen Radical Absorbance Capacity (ORAC), Ferric Reducing Ability of Plasma (FRAP) and Trolox Equivalent Antioxidant Capacity (TEAC). Moreover, the DPPH method without any extraction procedure (Q-DPPH) was used to evaluate the total antioxidant activity of the samples.

#### 2.12.1. DPPH Assays (Classical and Q-Versions)

The classical version of DPPH assay was performed as previously described [15], with slight modifications. An amount of 25 µL of extracts was mixed with 100 µL of MilliQ water and 125 µL of DPPH· working solution (100 µM using methanol as solvent) in a 96-well microplate. Absorbance at 515 nm was recorded for 30 min in a microplate reader.

The Q-DPPH assay was carried out by mixing 10 mg of barley samples with 1.6 mL DPPH· working solution (50 µM) prepared in either methanol (Q-DPPH_M) or methanol:water (50:50, *v*:*v*) (Q-DPPH_MW). A volume of 160 µL of methanol:water (50:50, *v*:*v*; pH = 2) was added for the calibration curve with Trolox to compensate for the solvent present in the sample reactions. The samples were incubated in an orbital shaker (750 rpm for 30 min) then centrifuged (10,000× *g* for 2 min). Absorbance was recorded at 515 nm in a microplate reader. The results were corrected for moisture and expressed as µg Trolox equivalents/100 g d.m.

#### 2.12.2. TEAC Assay

The classical version of this method was carried out according to Re et al. [16] with slight modifications. An aqueous ABTS solution (7 mM) was mixed with 2.45 mM K_2_O_8_S_2_ in a 1:1 (*v*/*v*) ratio (stock ABTS·^+^ solution). Stock ABTS·^+^ solution was diluted with phosphate buffer (75 mM, pH = 7.4) (ABTS·^+^ working solution), with an absorbance value of 0.70 ± 0.02 at 734 nm. Then, 20 µL of diluted samples was mixed with 200 µL ABTS·^+^ working solution in a 96-well microplate. The decay in absorbance at 730 nm was recorded over 30 min with a microplate reader. Trolox was used as the standard. The results were corrected for moisture and expressed µg Trolox equivalents/100 g d.m.

#### 2.12.3. ORAC Assay

ORAC assay was carried out as earlier described [17] with some modifications. Trolox standard curve (15-240 mM) and samples were diluted in phosphate buffer (10 mM, pH 7.4). Fluorescein (150 µL) was placed in a 96-well black microplate and then 25 µL of Trolox standard, sample or phosphate buffer (blank) was added. After incubation (37 °C, 3 min), AAPH solution was added to initiate the oxidation reaction. Fluorescence was monitored over 35 min with a microplate reader at excitation and emission wavelengths of 485 nm and 528 nm, respectively. The results were expressed as mmol Trolox Equivalents (TE)/g d.m.

#### 2.12.4. FRAP Assay

The assay was performed following the protocol reported by Vijayalakshmi and Ruckmani [18]. Results were expressed as mmol Fe equivalents/g d.m.

### 2.13. Glycemic Index

For the determination of glycemic index (GI) in sprouted barley, firstly, the content of available starch was measured using the total starch assay kit of Megazyme (K-TSTA 08/16). Afterwards, the in vitro starch hydrolysis rate was determined as described by Gularte and Rosell [19], with slight modifications. Samples containing 50 mg of available starch were dissolved in Tris-maleate buffer (0.1 M, pH = 6, 2 mL) and then 2 mL enzymatic solution containing porcine pancreatic α-amylase (460 U/mL) and amyloglucosidase (6.6 U/mL) were added. Aliquots were taken during the incubation period (150 min), were kept in boiling water for 5 min to stop the enzymatic reaction and cooled in ice. Then, a volume of 150 µL of absolute ethanol was added and the sample was centrifuged (4 °C, 10,000× *g* for 5 min). The pellet was washed with 150 µL ethanol:water (1:1, *v*/*v*) and the supernatants were pooled together and stored at 4 °C for the subsequent colorimetric analysis of reducing sugars using the GOPOD kit (Megazyme, Bray, Ireland). GI values were calculated from hydrolysis index (HI) values, as proposed by Granfeldt [20].

### 2.14. Colorimetric Characterizationof Germinated Barley Flours

Color was measured using a colorimeter (Minolta CM-2002, Osaka, Japan) with D65 as illuminant and 45/0 sensor. The instrument was calibrated with a white tile standard (L*593.97, a*520.88 and b*51.21). The CIE L*a*b* parameters were converted to Hue (arctan b*/a*), Chroma (a*^2^ + b*^2^)^1/2^ based in the CIELab (L*, a* and b*).

### 2.15. Sensory Analysis

Panelists (20–45 years old) with previous experience in sensory analysis were recruited from the Staff of the Agro-Technological Institute of Castilla y León. Panelists were instructed to rinse their mouths with water between samples. Samples were subjected to descriptive test. Colour, texture and flavour were evaluated for all the samples using a scale from 0 (disliked extremely) to 5 (liked extremely).

### 2.16. Statistical Analysis

Analytical determinations were carried out in duplicate for each replicate (three replicates were obtained for each germination condition). The equations obtained by RSM models were solved using the least squares (LS) method. Analysis of variance (ANOVA) was used to assess the reliability of the regression equations obtained and to determine the significance of all terms (linear, quadratic and interaction) of each factor at a probability level (*p*) of 0.05. The goodness of models to describe the variations of experimental data was also quantified by the coefficient of determination (*R*^2^). A quadratic model was used to evaluate the responses. A statistical analysis was performed using Statgraphic Centurion XVI (Rockville, MD, USA) software.

## 3. Results

The nutritional, bioactive and sensory properties of germinated barley were evaluated through different response variables using polynomic regression models (Table 2). ANOVA results reveal that the predictive models obtained for all response variables were statistically significant (*p* ≤ 0.05), with no significant values of lack-of-fit, indicating that the models satisfactorily predict the relationship between response variables and independent factors. Higher *R*^2^ values than 0.75 suggest an adequate fitting of the polynomial models to the experimental values. The germination yield was higher than 50% regardless of the conditions used during sprouting.

### 3.1. Effect of Germination on Proximate Composition of Sprouted Barley Flours

The polynomic regression equations (Table 2) and response surface tridimensional contour plots (Figure 2) obtained for proximate composition of germinated barley showed a significant higher effect of germination time than temperature on these nutritional parameters.

Fat levels of barley sprouts were significantly reduced compared to non-germinated grain and this decrease showed a linear time-dependent behavior (Figure 2a). Decrease of fat may be linked to depletion of the stored fat that is involved in grain catabolic activities required for protein synthesis in the growing plant [21].

Contrary results were obtained for protein content, which increased significantly after germination in all barley sprouts regardless of the germination conditions used (Figure 2b). The enhancement of protein content in seed sprouts might be attributed to the loss of carbohydrates during respiration that cause an apparent increase in other nutrients such as proteins, as has been previously reported in chia sprouts [22]. Higher temperature and longer time would produce greater loss in the sprouts dry weight and a more pronounced enhancement of protein levels. Moreover, there is a reawakening protein synthesis upon imbibition [23], which could also partially explain the increased protein content in sprouted barley. Again, the protein increase was only dependent on germination time, where a linear effect was observed.

The content of fiber in germinated barley decreased with respect to non-germinated grains (Figure 2c), that could be associated with the activation of hydrolytic enzymes, mainly at longer periods of germination. However, no significant differences were observed at different germination temperatures.

The carbohydrate content of sprouted barley decreased at low temperatures and short germination times and remained almost constant at the other temperatures (Figure 2d).

### 3.2. Effect of Germination on Fatty Acid Content of Sprouted Barley Flours

The content of saturated, unsaturated, monounsaturated and polyunsaturated fatty acids was also evaluated in germinated barley (Figure 3 and Table 2). Barley germinated at lower temperatures and shorter times exhibited levels of these compounds similar to non-germinated barley grain. Saturated fatty acids duplicated during sprouting, reaching values close to 37% at longer times of germination (Figure 3a) and the opposite behavior was observed for unsaturated fatty acids (Figure 3b). Polyunsaturated acids (Figure 3d) did not show significant differences compared to control barley. The values ranged from 59% to 63% with an insignificant decrease at higher temperatures and longer times of germination. Our results are in line with previous investigations showing that triglycerides that represent the major lipids storage form in cereals, are hydrolyzed by lipases to diglycerides, monoglycerides, and then to glycerol and fatty acids [24]. Hence, it seems that α-oxidation plays a minor role in sprouting grains while β-oxidation plays a key role with the aid of β-oxidase, yielding energy for metabolic pathways [25].

### 3.3. Effect of Germination on Vitamin Content of Sprouted Barley Flours

Vitamin content increased in barley after germination and it exhibited a significant linear dependence respect to temperature and time regardless of the vitamin studied. Higher levels of water-soluble vitamins (B_1_, B_2_ and C) were observed at longer germination times and higher temperatures (Figure 4). Vitamins B_1_ (Figure 4a) and C (Figure 4c) contents almost triplicated in barley germinated at longer time while vitamin B_2_ (Figure 4b) showed a six-fold increase in sprouted barley. Different authors have reported an increase of vitamins content during chickpea and barley germination as consequence of the biosynthesis undergone in the grain [26,27].

### 3.4. Effect of Germination on β-Glucan Content of Sprouted Barley Flours

The β-glucan content was determined in raw and germinated barley (Table 2 and Figure 5a). The results show that β-glucan levels in barley grains were close to 5 g/100 g d.m. and decreased to a short extent during germination for short times (reductions of 14%–22% with respect to the initial content after 84-86 h of germination), but longer germination times cause greater losses. The impact of germination time on β-glucan levels was stronger at lower temperatures. It has been reported that germination causes an increase of β-glucanases activity, which are the primary enzymes responsible for breaking down cereal endosperm cell walls during germination, thus reducing β-glucan levels [28,29]. Our results agree with those reported by Rimsten et al. [30], who showed slight losses in β-glucan levels in barley grains germinated for 96 h after steeping at 48 °C. On the contrary, larger reductions of β-glucan content (50% after 48–120 h of germination) were found in germinated barley [31]. These studies have shown that longer germination times caused higher β-glucan degradation, results in line with the ones found in the present study.

### 3.5. Effect of Germination on GABA Content of Sprouted Barley Flours

Non-sprouted barley showed a GABA content of around 54 mg/100 g d.m. that significantly increased during germination (Table 2, Figure 5b), reaching values between 81 and 186 mg/100 g d.m. in sprouted barley, depending on germination conditions. The increase of GABA content in barley sprouts can be attributed to the partial hydrolysis of storage proteins to oligopeptides and free amino acids that are used for seedlings growth during germination, followed by the activation of glutamate decarboxylase enzyme that converts glutamic acid to GABA [10,32]. GABA can also be synthesized from polyamines by diamine oxidase (DAO), whose activity has been reported to increase during grain germination [33]. The enhancement of GABA content during cereal germination has been also observed by other authors in rice, barley and wheat [9,10,34].

### 3.6. Effect of Germination on Total Phenolic Content of Sprouted Barley Flours

One of the main advantages of germination in controlled conditions is the possibility of enhancing the antioxidant properties of grains by increasing phenolics contents.

The total phenolic contents (TPCs) increased significantly in barley after germination compared to the control, regardless of the conditions used during the process (Table 2 and Figure 5c). TPC levels ranged from 4.4 to 9.5 μmol GA Eq/g d.m. in sprouted barley. Germination time showed a significant linear impact on TPC content, especially at higher temperatures. The TPC levels in raw barley were in agreement with the values reported by Zhao et al. [35], who reported contents between 3.07 and 4.48 μmol GA Eq/g d.m. in different barley varieties. The increased levels of these bioactive compounds during barley sprouting is associated to structural changes of barley matrix that cause the release of phenolic compounds bound to cellular structures, and also to glycosylation reactions during germination, enhancing the extraction of free phenolic acids [36,37].

### 3.7. Effect of Germination on Free and Bound Phenolic Compounds Profile of Sprouted Barley Flours

Table 3 shows the content of free and bound phenolic compounds in barley seed and sprouts obtained at different conditions. Free phenolic fraction was mainly constituted of flavanols such as catechin and its dimeric form procyanidin B, while hydroxycinnamic acids, namely ferulic acid, 4-coumaric acid and caffeic acid, were the most abundant bound phenols. Non-germinated barley seeds contained a high concentration of procyanidin B (12.44 ± 0.80 µg/mg of extract) and catechin (0.99 ± 0.16 µg per mg of extract) in the soluble fraction (free phenolics), while the insoluble fraction (bound phenolics) was abundant in ferulic acid (45.06 ± 3.09 µg/mg of extract), 4-coumaric acid, (2.34 ± 0.40 µg/mg of extract), procyanidin B (1.06 ± 0.14 µg per mg of extract) and 4-hydroxybenzoic acid (0.45 ± 0.01 µg/mg of extract). The results found in this study are in agreement with those previously reported in barley [38,39].

The content of ferulic acid in the bound fraction increased during sprouting as the germination time augmented at higher temperatures (Table 2 and Figure 6a), which might be attributed to *de novo* synthesis in the embryonic axis of sprouted grain [40]. Our results align with those previously observed in wheat by other authors [41,42], who found an increase of ferulic acid levels after germination for 24–96 h. As expected, the content of procyanidin B decreased during germination (Figure 6b), followed by a rise in catechin content. The decrease of procyanidin B levels might be associated with its degradation to individual catechin monomers to protect the seed at this very vulnerable growth stage.

### 3.8. Effect of Germination on Antioxidant Activity of Sprouted Barley Flours

Total antioxidant activity was measured as a comprehensive assessment through the evaluation of different antioxidant parameters (Table 2 and Figure 7). The germination process produced an enhancement of the antioxidant activity, which was mainly time-dependent, exhibiting the sprouts produced at longer germination times the highest antioxidant activities, regardless of the assay used for evaluation (FRAP, ORAC, DPPH, TEAC and Q-DPPH). The increase in antioxidant potential in sprouted barley could be attributed to the release of phenolic compounds from the cell wall components during germination process, compounds that can act as hydrogen or electron donors [43]. Since DPPH scavenging activity was also evaluated without a previous extraction step, this antioxidant activity in sprouted barley cannot be solely attributed to the increase of free phenolic compounds during germination. These findings suggest that other antioxidant compounds present in barley sprouts, such as vitamin C, also contribute to its high antioxidant potential.

### 3.9. Effect of Germination on Glycemic Index (GI) of Sprouted Barley Flours

The GI indicates the rate at which 50 g of carbohydrate in a particular food is absorbed into the blood stream as blood sugar and ranges from 1 to 100. Glucose is used as reference food and is rated 100. Foods can be classified by their GI, according to Gordillo-Bastidas et al. [44], as follows: (i) High GI foods (>70), (ii) Medium GI foods (between 56–69) and iii) Low GI foods (≤55).

GI was evaluated in all raw and sprouted barley samples (Table 2, Figure 8). As germination proceeded, GI increased significantly in barley as consequence of the release of reducing sugars. In fact, studies performed in barley and brown rice found an enhancement of reducing sugars levels during germination due to hydrolysis of starch by α-amylase and β-amylase enzymes [31,45].

### 3.10. Effect of Germination on the Colour and Sensory Properties of Sprouted Barley Flours

Two phenomena were observed when the colour of germinated barley flours was compared with that of raw barley flour (Table 2, Figure 9 and Figure 10). Germination decreased the brightness (L*) of barley flour. However, redness (a*) and yellowness (b*) increased as the germination proceed (Table 2). These results can be attributed to the starch and protein hydrolysis during germination and, subsequently, the generation of Maillard reaction compounds during drying treatment. Moreover, during the soaking period, there is a migration of bran pigments into the water and into the endosperm. Moreover, heating treatments might affect the brown barley colour. These results are in agreement with the results observed by the sensory panel (Table 2 and Figure 11) who observed an increase in colour change associated with the loss of brightness and increased flavor intensity in germinated barley. Texture was also modified, as an increasing coarseness of the powder with germination temperature was observed.

### 3.11. Multivariate Optimization of Germination Conditions to Obtain High-Quality Germinated Barley Flours

The desirability function was used to select the optimal germination conditions to obtain the highest predicted values for nutritional and bioactive compounds as well as for antioxidant activity and the lowest values for saturated fatty acid content and glycemic index. Based on this criterion, the most suitable germination conditions were temperature and time of 16 °C and 3.53 days, respectively, conditions that provide an overall desirability value of 0.44 (Figure 12).

## 4. Conclusions

This study demonstrates that germination is a promising process for developing novel nutritive and functional flours from barley with improved quality features. An increase in germination time caused an enhancement of the nutritional and bioactive potential of sprouted barley flour, but also an increase of carbohydrates hydrolysis and, consequently, a higher glycemic index. Since the aim of this study was to produce improved barley flours from the nutritional and functional point of view, the use of mild temperatures (16 °C) and short times (3.53 days) could be promising to produce nutritive barley flours with high antioxidant properties and a low glycemic index, which may be beneficial for consumers´ health.

## Figures and Tables

**Figure 1 foods-09-00296-f001:**
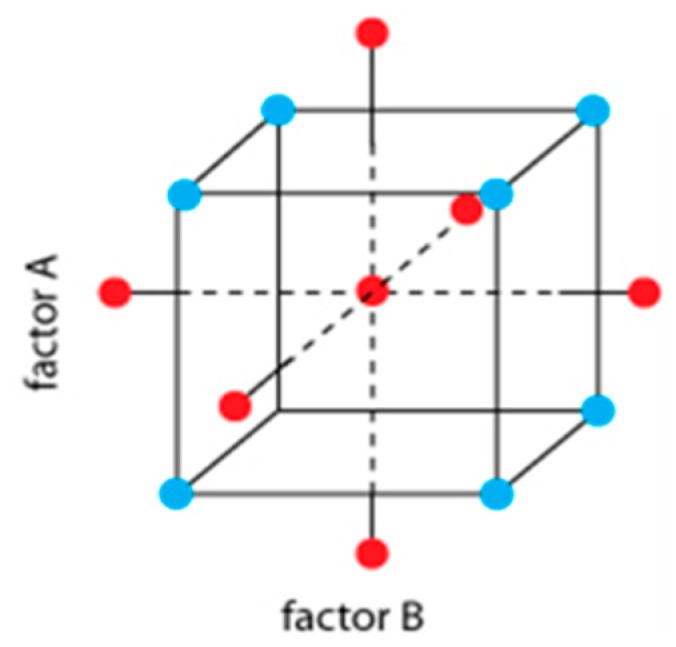
Response surface central composite design with two independent variables (factors), three center points and two star points.

**Figure 2 foods-09-00296-f002:**
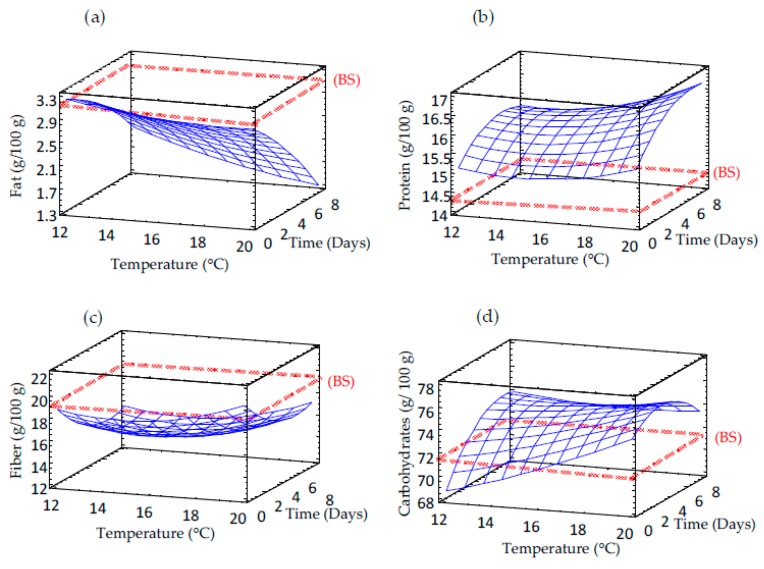
Response surface plots for proximal composition in germinated barley flour as a function of two independent variables (time and temperature). (**a**) Fat content; (**b**) Protein content; (**c**) Fiber content; (**d**) Carbohydrate content. BS: barley seed flour (non-germinated).

**Figure 3 foods-09-00296-f003:**
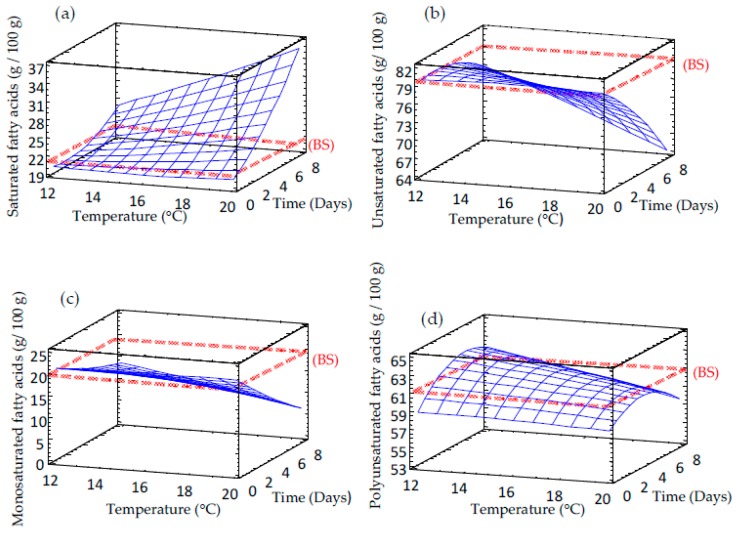
Response surface plots for fatty acid content in germinated barley flour as a function of two independent variables (time and temperature). (**a**) Saturated fatty acids; (**b**) Unsaturated fatty acids; (**c**) Monounsaturated fatty acids; (**d**) Polyunsaturated fatty acids. BS: barley seed flour (non-germinated).

**Figure 4 foods-09-00296-f004:**
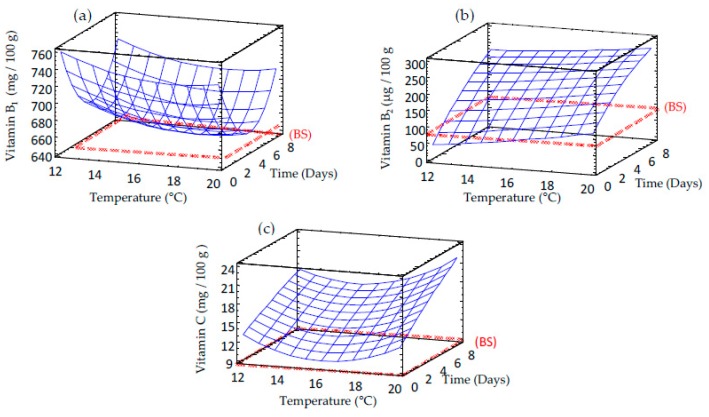
Response surface plots for vitamin content in germinated barley flour as a function of two independent variables (time and temperature). (**a**) Vitamin B_1_; (**b**) Vitamin B_2_ (**c**) Vitamin C. BS: barley seed flour (non-germinated).

**Figure 5 foods-09-00296-f005:**
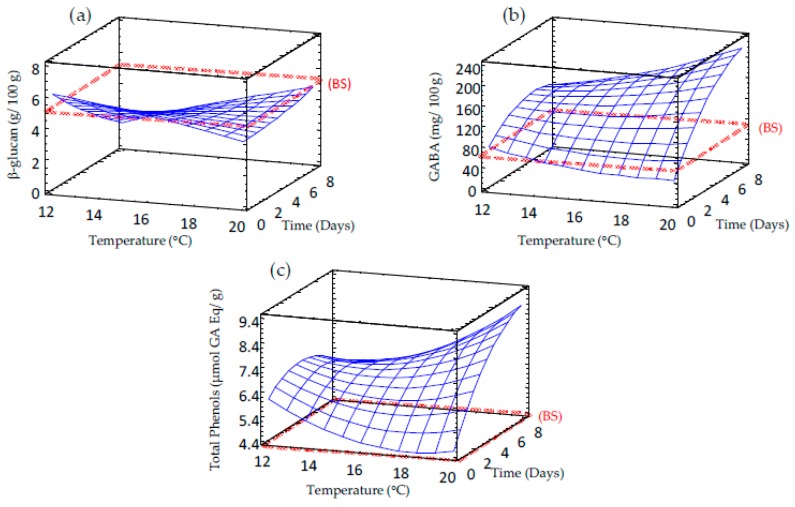
Response surface plots for bioactive compounds content in germinated barley flour as a function of two independent variables (time and temperature). (**a**) β-glucan; (**b**) GABA; (**c**) TPC. BS: barley seed flour (non-germinated).

**Figure 6 foods-09-00296-f006:**
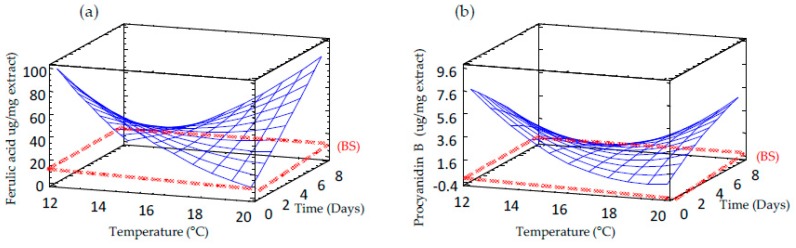
Response surface plots for predominant free and bound phenolic compounds in germinated barley flour as a function of two independent variables (time and temperature). (**a**) Ferulic acid; (**b**) Procyanidin B. BS: non-germinated barley flour.

**Figure 7 foods-09-00296-f007:**
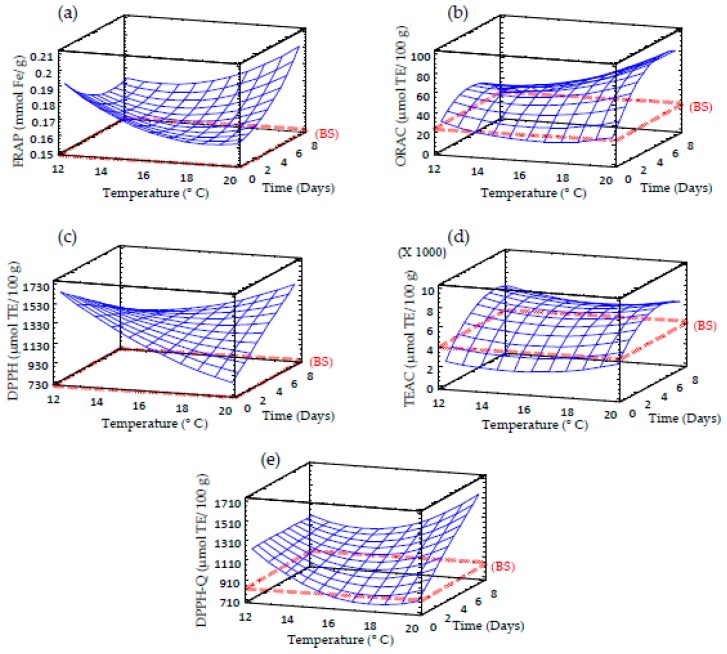
Response surface plots for antioxidant activity in germinated barley flour as a function of two independent variables (time and temperature). (**a**) FRAP; (**b**) ORAC; (**c**). DPPH; (**d**) TEAC; (**e**) DPPH-Q. BS: barley seed flour (non-germinated).

**Figure 8 foods-09-00296-f008:**
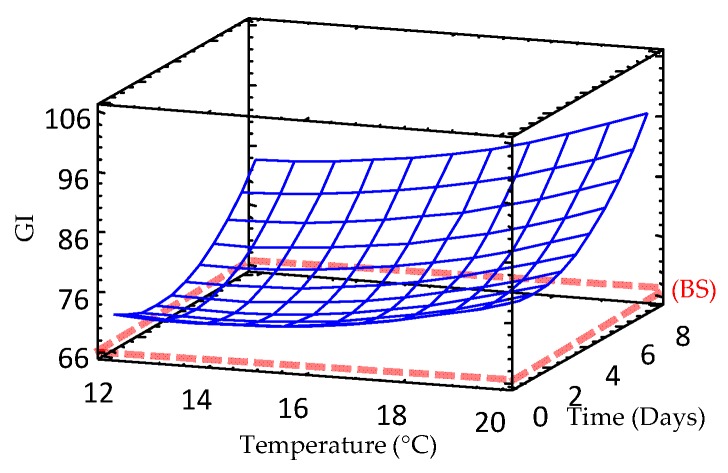
Response surface plots for glycemic index in germinated barley flour as a function of two independent variables (time and temperature). BS: barley seed flour (non-germinated).

**Figure 9 foods-09-00296-f009:**
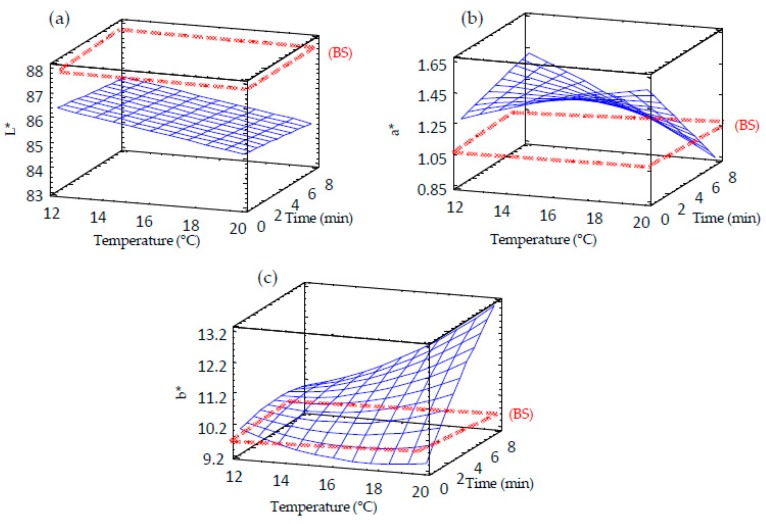
Response surface plots for color in germinated barley flour as a function of two independent variables (time and temperature). (**a**) L* value; (**b**) a* value; (**c**) b* value. BS: barley seed flour (non-germinated).

**Figure 10 foods-09-00296-f010:**
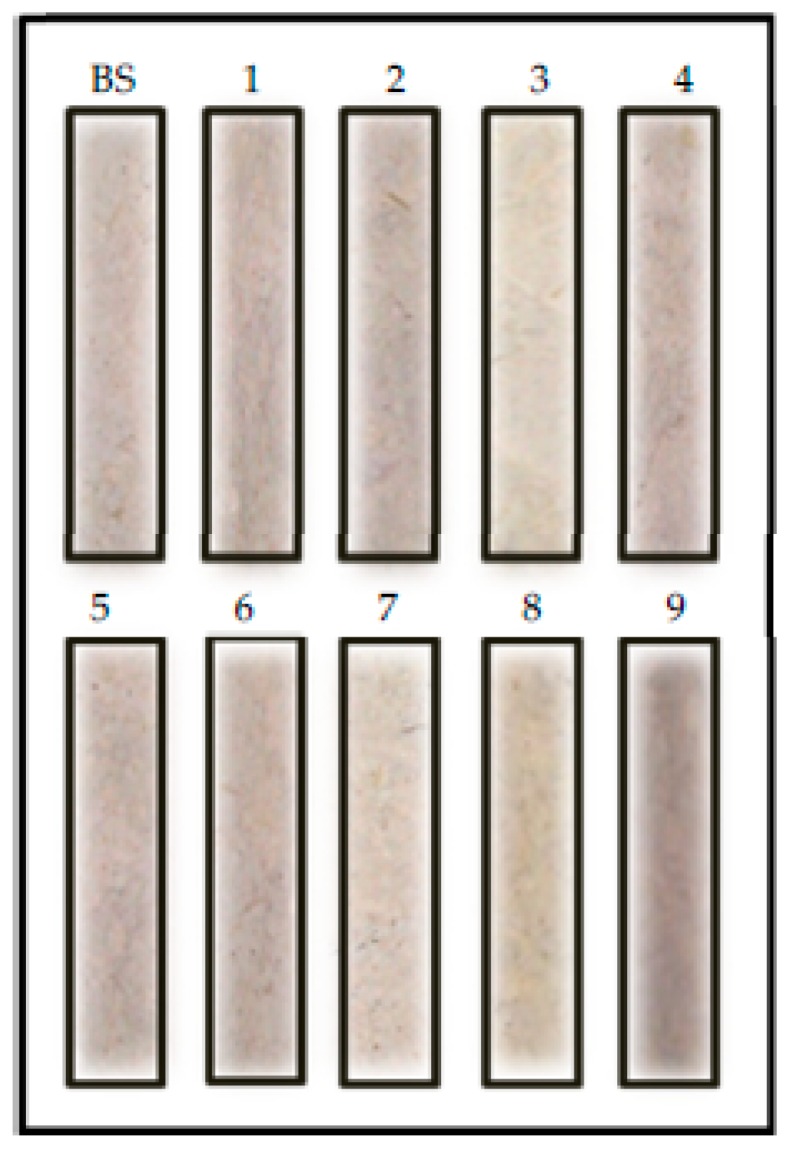
Colour of flours from germinated barley flour obtained at different conditions of time and temperature. BS: barley seed flour (non-germinated). Codes 1 to 9 corresponded to the germination experiments performed according to RSM design (Table 1).

**Figure 11 foods-09-00296-f011:**
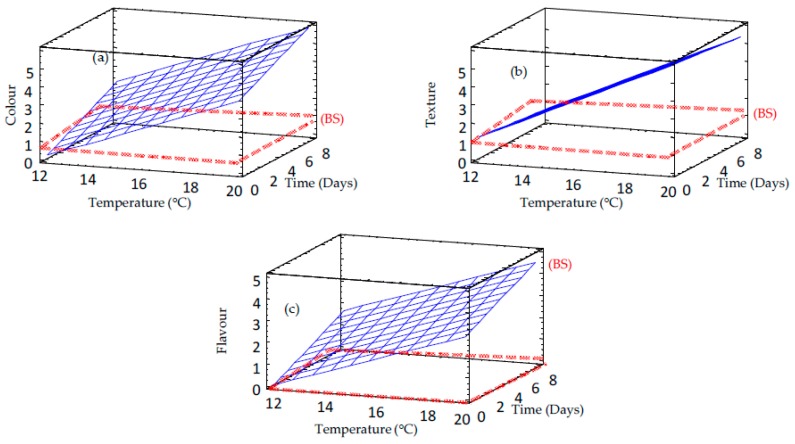
Response surface plots for sensory properties of germinated barley flour as a function of two independent variables (time and temperature). (**a**) Colour; (**b**) Texture; (**c**) Flavour. BS: non-germinated barley flour.

**Figure 12 foods-09-00296-f012:**
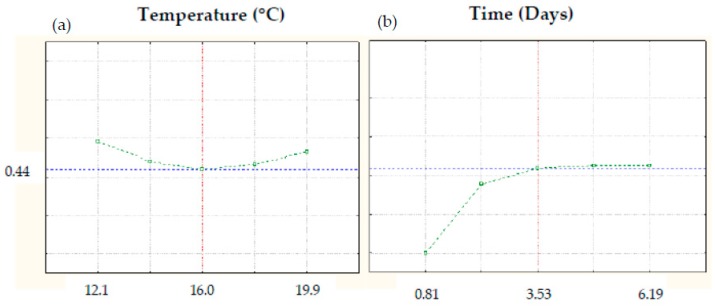
Optimal germination conditions determined by desirability function. (**a**) Optimal temperature; (**b**) Optimal time.

**Table 1 foods-09-00296-t001:** Independent variables (factors) of the response surface central composite design.

Factor A Temperature (°C)	Factor B Time (Days)	Experiments (Codes)
12.1	3.58	1
13.2	1.60	2
13.2	5.52	3
16.0	0.81	4
16.0	3.48	5*
16.0	6.19	6
18.8	1.60	7
18.8	5.52	8
19.9	3.60	9

*Center point. Three replicate experiments were performed for the center point.

**Table 2 foods-09-00296-t002:** Polynomic equations and optimum values for the dependent variables studied in the RSM design.

Response Variables	Model Equations	Optimum Temperature (Tp)	Optimum Time (Tm)
Fat	4.984 – 0.214 × TP + 0.022 × TM + 0.006 × TP^2^ – 0.008 × TM × TP – 0.009 × TP^2^	12.1	0.81
Protein	19.066 – 0.551 × TM + 0.167 × TP + 0.019 × TM^2^ + 0.010 × TM × TP – 0.026 × TP^2^	19.9	6.19
Fiber	35.541 – 2.362 × TP – 1.176 × TM + 0.080 × TP^2^ + 0.00004 × TM × TP + 0.098 × TP^2^	19.9	0.81
Carbohydrates	65.948 – 0.075 × TM + 2.804 × TP + 0.025 × TM^2^ – 0.098 × TM × TP – 0.121 × TP^2^	12.1	0.81
Saturated fatty acids	24.444 – 0.587 × TM – 2.426 × TP + 0.019 × TM^2^ + 0.179 × TP × TM + 0.010 × TP^2^	19.9	6.19
Unsaturated fatty acids	75.556 + 0.587 × TM + 2.426 × TP – 0.019 × TM^2^ – 0.179 × TM × TP – 0.100 × TP^2^	12.1	0.81
Monounsat. fatty acids	17.847 + 0.398 × TM + 0.288 × TP – 0.010 × TP^2^ – 0.123 × TM × TP + 0.036 × TP^2^	14.3	0.81
Polyunsat. fatty acids	57.747 + 0.184 × TM + 2.141 × TP – 0.008 × TP^2^ – 0.056 × TM × TP – 0.136 × TP^2^	12.1	5.4
Vitamin C	61.315 – 6.418 × TP + 0.101 × TM + 0.110 × TP^2^ + 0.060 × TM × TP – 0.020 × TP^2^	19.9	6.19
Vitamin B_1_	1003.83 – 30.743×TM – 37.626×TP + 0.867 × TM^2^ + 0.059 × TP × TM + 4.207 × TM^2^	12.1	0.81
Vitamin B_2_	4.880 – 0.329 × TP + 39.175 × TM + 0.290 × TP^2^ – 0.460 × TM × TP – 1.271 × TP^2^	19.9	6.19
β-glucan	6.886 + 0.052 × TP – 1.68 × TM – 0.009 × TP^2^ + 0.082 × TP × TM + 0.014 × TP^2^	12.1	0.81
GABA	311.39 – 28.559 × TM – 4.590 × TP + 0.759 × TM^2^ + 2.101 × TP × TM – 1.924 × TM^2^	19.9	6.19
TPC	367.582 – 31.560 × TP – 8.571 × TM + 0.874 × TM^2^ + 1.297×TP × TM – 0.929 × TP^2^	19.9	6.19
Bound ferulic acid	423.975 – 36.487 × TP – 47.023 × TM + 0.784 × TP^2^ + 2.769 × TM × TP + 0.212 × TP^2^	12.1	0.81
Free procyanidin	53.593 – 5.589 × TP – 2.911 × TM + 0.147 × TP^2^ + 0.181 × TM × TP – 0.025 × TP^2^	12.1	0.81
FRAP	0.407 – 0.027 × TP – 0.017 × TM + 0.0007 × TP^2^ + 0.0009 × TM × TP + 0.0005 × TP^2^	19.9	6.19
ORAC	205.967 – 23.180 × TM – 0.708 × TP + 0.709 × TM^2^ + 0.923 × TM × TP – 1.361 × TP^2^	19.9	6.19
DPPH	4274.09 – 291.532 × TP – 284.032 × TM + 5.985 × TP^2^ + 18.461 × TM × TP – 0.231 × TP^2^	12.1	0.81
TEAC	15296.3 – 1782.21 × TM + 1758.61 × TP + 59.311 × TM^2^ – 21.386 × TP × TM − 119.929 × TP^2^	19.9	6.19
DPPH-Q *	5252.4 – 507.735 × TP – 136.6 × TM + 14.287 × TP^2^ + 11.670 × TM × TP – 0.503 × TM^2^	19.9	6.19
L*	100.793 – 1.859 × TM + 0.478 × TP + 0.054 × TM^2^ – 0.003 × TM × TP – 0.069 × TP^2^	12.1	3.2
a*	0.455 + 0.088 × TM + 0.182 × TP − 0.002 × TM^2^ – 0.013 × TM × TP – 0.0004 × TP^2^	19.9	0.81
b*	16.906 – 0.874 × TM – 0.564 × TP + 0.025 × TM^2^ + 0.057 × TP × TM – 0.014 × TP^2^	12.1	0.81
Glycemic index	105.067 – 4.918 × TM – 3.373 × TP + 0.180 × TM^2^ + 0.093 × TP × TM + 0.471 × TP^2^	19.9	6.19
Color	(−) 0.322 + 0.0469 × TM + 0.075 × TP	12.1	0.81
Texture	0.268 + 0.039 × TM − 0.011 × TP	12.1	0.81
Flavour	(−) 0.365 + 0.047 × TM + 0.085 × TP	19.9	6.19

FRAP: Ferric Reducing Ability of Plasma; ORAC: Oxygen Radical Absorbance Capacity; DPPH: 2,2-difenil-1-picrylhydrazyl Radical Scavenging Activity; TEAC:Trolox Equivalent Antioxidant Capacity; DPPH-Q: Direct DPPH method (without extraction); L*: Luminosity; a*: CIELAB a* parameter; b*: CIELAB b* parameter.

**Table 3 foods-09-00296-t003:** Content of free and bound phenolic compounds (μg/mg extract) in germinated barley flour as a function of two independent variables (time and temperature).

Fraction	Germination Experiments	Catechin	Protocatechuic Acid	Diferulic Acid	4-hydroxybenzoic Acid	Phloridzin	Ferulic Acid	Caffeic Acid	4-coumaric Acid	Procyanidin B
**Free**	**BS**	0.99 ± 0.16 ^cB^	0.01 ± 0.02 ^aA^	2.22 ± 0.31 ^dC^	ND ^aA^	0.01 ± 0.00 ^aA^	ND ^aA^	ND ^aA^	ND ^aA^	12.44 ± 0.80 ^cD^
	**1**	0.20 ± 0.04 ^bB^	0.02 ± 0.00 ^bA^	0.76 ± 0.32 ^cC^	ND ^aA^	0.02 ± 0.01 ^aA^	ND ^aA^	ND ^aA^	ND ^aA^	1.94 ± 0.56 ^abB^
	**2**	0.02 ± 0.00 ^aA^	0.02 ± 0.00 ^bA^	0.36 ± 0.19 ^bB^	ND ^aA^	0.02 ± 0.01 ^aA^	ND ^aA^	ND ^aA^	ND ^aA^	3.26 ± 1.41 ^bC^
	**3**	0.53 ± 0.40 ^bcB^	0.01 ± 0.02 ^aA^	1.19 ± 0.02 ^eB^	ND ^aA^	0.03 ± 0.01 ^aA^	ND ^aA^	ND ^aA^	ND ^aA^	1.40 ± 0.57 ^abB^
	**4**	0.12 ± 0.12 ^aA^	0.03 ± 0.00 ^bA^	1.06 ± 0.07 ^deB^	ND ^aA^	0.01 ± 0.00 ^aA^	ND ^aA^	ND ^aA^	ND ^aA^	1.06 ± 0.01 ^aB^
	**5**	0.18 ± 0.02 ^bB^	0.02 ± 0.00 ^bA^	1.61 ± 0.84 ^defC^	ND ^aA^	0.02 ± 0.01 ^aA^	ND ^aA^	ND ^aA^	ND ^aA^	1.68 ± 1.11 ^abC^
	**6**	0.67 ± 0.71 ^bcAB^	0.03 ± 0.00 ^bA^	1.90 ± 0.82 ^efB^	ND ^aA^	0.01 ± 0.00 ^aA^	ND ^aA^	ND ^aA^	ND ^aA^	2.79 ± 2.23 ^aB^
	**7**	0.08 ± 0.08 ^aA^	0.03 ± 0.00 ^bA^	0.65 ± 0.00 ^cB^	ND ^aA^	0.01 ± 0.00 ^aA^	ND ^aA^	ND ^aA^	ND ^aA^	0.49 ± 0.70 ^aAB^
	**8**	0.87 ± 0.75 ^bcB^	0.02 ± 0.00 ^bA^	1.03 ± 0.10 ^cdB^	ND ^aA^	0.01 ± 0.00 ^aA^	ND ^aA^	ND ^aA^	ND ^aA^	2.68 ± 2.05 ^abB^
	**9**	0.57 ± 0.36 ^bcB^	0.03 ± 0.00 ^bA^	0.99 ± 0.02 ^dB^	ND ^aA^	0.02 ± 0.00 ^aA^	ND ^aA^	ND ^aA^	ND ^aA^	6.72 ± 2.87 ^bC^
**Bound**	**BS**	ND ^aA^	0.07 ± 0.05 ^bcB^	0.26 ± 0.05 ^bD^	0.45 ± 0.01 ^dE^	0.02 ± 0.01 ^aA^	45.06 ± 3.09 ^dH^	0.16 ± 0.01 ^bC^	2.34 ± 0.40 ^cdG^	1.07 ± 0.14 ^aF^
	**1**	ND ^aA^	0.10 ± 0.00 ^cB^	0.09 ± 0.00 ^aB^	0.31 ± 0.02 ^cC^	0.01 ± 0.00 ^aA^	36.45 ± 2.44 ^cF^	0.70 ± 0.09 ^dC^	2.46 ± 0.30 ^cdE^	1.22 ± 0.18 ^aD^
	**2**	0.02 ± 0.03 ^aA^	0.03 ± 0.01 ^abA^	0.09 ± 0.00 ^aB^	0.21 ± 0.04 ^bC^	0.01 ± 0.00 ^aA^	50.27 ± 9.18 ^deF^	0.13 ± 0.01 ^bB^	2.32 ± 0.43 ^cdE^	1.07 ± 0.33 ^aD^
	**3**	ND ^aA^	0.08 ± 0.06 ^bcB^	0.19 ± 0.15 ^abBC^	0.22 ± 0.02 ^bC^	0.02 ± 0.01 ^aA^	31.15 ± 1.89 ^cE^	0.33 ± 0.19 ^bcC^	1.56 ± 0.27 ^bcD^	1.23 ± 0.29 ^aD^
	**4**	ND ^aA^	0.05 ± 0.00 ^bB^	0.19 ± 0.02 ^bC^	0.27 ± 0.09 ^bcC^	0.02 ± 0.01 ^aA^	31.25 ± 5.16 ^cF^	0.22 ± 0.01 ^cC^	1.92 ± 0.19 ^cd.E^	0.90 ± 0.03 ^aD^
	**5**	ND ^aA^	0.06 ± 0.01 ^bB^	1.18 ± 0.1 ^eD^	0.15 ± 0.08 ^bB^	0.01 ± 0.01 ^aA^	23.82 ± 1.02 ^bF^	0.67 ± 0.01 ^dC^	1.40 ± 0.03 ^bE^	0.66 ± 0.94 ^aBCD^
	**6**	ND ^aA^	0.08 ± 0.02 ^bc.B^	1.52 ± 0.70 ^defCD^	0.22 ± 0.18 ^bcB^	0.01 ± 0.00 ^aA^	28.40 ± 17.43 ^bcdE^	0.68 ± 0.41 ^cdC^	2.01 ± 1.06 ^bcdD^	0.89 ± 0.06 ^aC^
	**7**	ND ^aA^	0.04 ± 0.03 ^abA^	1.03 ± 0.11 ^dC^	0.12 ± 0.07 ^bB^	0.01 ± 0.01 ^aA^	31.27 ± 9.38 ^cdE^	0.26 ± 0.10 ^cB^	1.79 ± 0.57 ^bcD^	1.08 ± 0.14 ^aC^
	**8**	ND ^aA^	0.15 ± 0.07 ^cB^	1.21 ± 1.12 ^bcdeCD^	0.41 ± 0.01 ^dC^	0.02 ± 0.01 ^aA^	72.95 ± 12.73 ^eF^	1.21 ± 0.21 ^eD^	2.60 ± 0.51 ^cdE^	0.99 ± 0.07 ^aD^
	**9**	ND ^aA^	0.13 ± 0.03 ^c^	2.88 ± 0.42 ^fE^	0.32 ± 0.09 ^cC^	0.01 ± 0.01 ^aA^	43.89 ± 6.59 ^dF^	0.98 ± 0.05 ^eD^	3.28 ± 0.85 ^dE^	1.01 ± 0.07 ^aD^

The results are the mean ± std (*n* = 3). Different superscript letters (small) within a column indicate significant differences (*p* ≤ 0.05) among germination treatments. Different superscript letters (capital) within a row indicate significant differences among individual phenolic compounds. Codes 1–9 corresponded to the germination experiments performed according to the RSM design (Table 1). BS: Barley seed flour (non-germinated). ND: not detected.

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
