# Peer review of "Sprouted Barley Flour as a Nutritious and Functional Ingredient"

_foods, 2020, doi:10.3390/foods9030296_

Round 1
Reviewer 1 Report
Detailed recommendation:
Abstract: please add more data to the abstract.
Introduction: please add more information about functional ingredients.
Figure 6: the presentation of results is illegible, please change it
In my opinion, it is more readable to present the results in the tables.
Are the results statistically significantly different? Please show it.
Author Response
Comment: Abstract: please add more data to the abstract.
Author’s response: Abstract has been modified in order to include the increases of bioactive compounds of sprouts obtained in the optimal germination conditions determined by RSM.
Comment: Introduction: please add more information about functional ingredients.
Author’s response: A paragraph regarding functional ingredients has been added, following the reviewer’s recommendation (p. 2, lines 61-67). As a consequence, a new reference has been added (Ref. 8, Vicentini et al., 2016).
Comment: Figure 6: the presentation of results is illegible, please change it
Author’s response: Figure 6 has been transformed into a table (Table 6) to facilitate the understanding of the results, as the reviewer recommended.
Comment: In my opinion, it is more readable to present the results in the tables.
Author’s response: Results from Figure 6 has been presented in a Table to make easier its understanding. The other results showed in the manuscript correspond to response surface plots, which are the graphical representation of predictive models obtained for each response variable (polynomial equations collected in Table 2). Therefore, we consider that the presentation of these results in tables would make difficult the understanding of the predictive models.
Comment: Are the results statistically significantly different? Please show it.
Author’s response The results presented are obtained from mathematical modeling performed using Response Surface Methodology (RSM). RSM allows to obtaining predictive models for explaining variations in the experimental values of response variables (fat, protein, fibre, etc…) caused by the independent factors (germination time and temperature). ANOVA has been performed to evaluate the significance of the polynomial models obtained for each response variable. Moreover, the coefficient of determination (R2) was used to estimate the adequacy of the model. ANOVA results showed that models developed for all response variables were statistically significant (p£0.05), with no significant lack of fit, indicating that the models adequately described the relationship between response and independent variables. The values of R2 >0.75 suggest a satisfactory fitting of the polynomial equations to the experimental data. All this information has been included in the revised manuscript (p.7-8, lines 244-249 and 254-258).
Reviewer 2 Report
The present study deals with the optimization of germination time and temperature for production novel nutritive and functional barley flour by using Response Surface Methodology (RSM).
Abstract
Line 23 - not only ORAC test was used
Because of B-glucan nutritional value, degree of B-glucan degradation at the optimal germination process (16oC, 3.5 days) should be added to the Abstract
Introduction
Line 49 – Add some newly Reference instead Ref 5 or 6 – for example Šimić, G. et al. Distribution of β-Glucan, Phenolic Acids, and Proteins as Functional Phytonutrients of Hull-Less Barley Grain. Foods 2019, 8, 680; doi:10.3390/foods8120680
Line 71 – Specify what “functional baking products” are about, whether gluten-free products or another products
Materials and Methods
Uniformly indicate the number of repetitions for all methods/analysis.
Line 82 - Make clear that H13 is a hull-less barley variety. As no agro-technical/climate conditions for barley growing during 2017/2018 are given, the variety pedigree is irrelevant for this study, stating that it is a commercial cultivar with a note from whom the sample was obtained is sufficient.
Before Germination paragraph specify the characteristics of the raw barley flour (BS) - what type of mill was used for grinding and consequently which type of flour was obtained, wholegrain or endosperm type of flour?
Paragraphs 2.6, 2.7, 2.9 and 2.11.
In describing all HPLC and LC-MS analysis, the separation conditions (detector, column, temperature, flow, solvents) should be uniformly presented.
Paragraphs 2.12.
The manuscript is too long, reduce the number of methods for assessment of AOC activities on two most effective tests (ORCAD +DPPH, or FRAP+DPPH….), without losing the manuscript value.
Paragraphs 2.13. Too long description.
Results
Line 263 - Results and Discussion
Comparing the results with other authors, specify which sprouting food crops (except barley) were used in the studies cited, for example Line 322, Ref 29-30, ….authors have reported an increase in vitamin C content during germination of soybean and chickpea seed; Line 346 Ref 9 – brown rice, Ref 37 – wild rice, etc.
Line 342 - 81-186 mg/100
Lines 371-375 - ug/mg extract
Lines 408-49 - comma after (>70)
Line 409 - Space (between 56-69)
Line 439 - Figure 11 Relocated letters and numbers above bars, BG or BS ?
References
Line 538 - Add year of publication in Ref. 23
Given the presence of enough references published in the last 10 years, reduce the number of older references by at least 20%
Author Response
Comment: not only ORAC test was used
Author’s response: As the reviewer indicated, different assays were used to evaluate the antioxidant activity of barley sprouts obtained at different germination conditions. However, only the antioxidant activity determined by ORAC increased from 2 to 4-fold in sprouts compared to raw grains, and therefore we have highlighted this result in the abstract.
Comment: Because of B-glucan nutritional value, degree of B-glucan degradation at the optimal germination process (16oC, 3.5 days) should be added to the Abstract
Author’s response: Sprouts obtained in optimal germination conditions retained 87% of the initial b-glucan content present in the non-sprouted grain. Moreover, these sprouts exhibited levels of ascorbic acid, riboflavin, phenolic compounds and GABA 1.4-2.5-fold higher than raw grain. This information has been included in the abstract (lines 32-35) of the revised manuscript, as suggested by the reviewer.
Comment: Line 49 – Add some newly Reference instead Ref 5 or 6 – for example Šimić, G. et al. Distribution of β-Glucan, Phenolic Acids, and Proteins as Functional Phytonutrients of Hull-Less Barley Grain. Foods 2019, 8, 680; doi:10.3390/foods8120680
Author’s response: References 5 and 6 have been replaced by that suggested by the reviewer (Šimić et al., 2019, reference 5 of revised manuscript).
Comment: Line 71 – Specify what “functional baking products” are about, whether gluten-free products or another products
Author’s response: Sprouted barley can be used to obtain functional gluten-containing products such as breads, cookies, cakes, muffins, etc. This information has been included in the revised manuscript (p.2, line 78).
Comment: Uniformly indicate the number of repetitions for all methods/analysis.
Author’s response. Analytical determinations were performed in duplicate for each sample replicate, taking into account that three replicates (sprouts samples) were obtained for each germination condition. This information has been included in the revised manuscript (p.7, lines 262-263) and removed from the other sections.
Comment: Line 82 - Make clear that H13 is a hull-less barley variety. As no agro-technical/climate conditions for barley growing during 2017/2018 are given, the variety pedigree is irrelevant for this study, stating that it is a commercial cultivar with a note from whom the sample was obtained is sufficient.
Author’s response. We have modified the paragraph including the type of variety (hull less) and the place where it was obtained, removing the data of growth, according to the reviewer’s recommendation (p. 2, lines 89-91).
Comment: Before Germination paragraph specify the characteristics of the raw barley flour (BS) - what type of mill was used for grinding and consequently which type of flour was obtained, wholegrain or endosperm type of flour?
Author’s response. We have included the information requested by the reviewer in the revised manuscript (p. 2-3, lines 91-96)
Comment: Paragraphs 2.6, 2.7, 2.9 and 2.11. In describing all HPLC and LC-MS analysis, the separation conditions (detector, column, temperature, flow, solvents) should be uniformly presented.
Author’s response. HPLC analysis of vitamins B1 and B2 (section 2.7) and GABA (section 2.9) has been briefly and uniformly described since references in which the methods are described in detail are included. The paragraph regarding the quantification of free and bound phenolics by LC-MS has been shortened by including a reference (p. 5, lines 179-181), in order to standardize the description of the method with sections 2.7 and 2.9, as the reviewer recommended.
Comment: Paragraphs 2.12. The manuscript is too long, reduce the number of methods for assessment of AOC activities on two most effective tests (ORCAD +DPPH, or FRAP+DPPH….), without losing the manuscript value.
Author’s response. According to the reviewer’s recommended, we have shortened the description of the methods used for antioxidant activity determination as much as possible, without remove important data for the readers.
Comment: Paragraphs 2.13. Too long description.
Author’s response. Description of glycemic index (section 2.13) has been shortened, according to the reviewer’s comment (p. 6, lines 221-232).
Comment: Line 263 - Results and Discussion. Comparing the results with other authors, specify which sprouting food crops (except barley) were used in the studies cited, for example Line 322, Ref 29-30, ….authors have reported an increase in vitamin C content during germination of soybean and chickpea seed; Line 346 Ref 9 – brown rice, Ref 37 – wild rice, etc.
Author’s response: According to the reviewer’s suggestion, we have specified which sprouted crops were used in the cited studies (lines 275-276, 315, 370, 387, 416).
Comment: Line 371-375 - ug/mg extract
Author’s response: Done (p.11, line 365-369).
Comment: Line 342 - 81-186 mg/100
Author’s response: Done (p.10, line 335).
Comment: Lines 408-49 - comma after (>70)
Author’s response: Done (p.12, line 413).
Comment: Lines 409 - Space (between 56-69)
Author’s response: Done (p.12, line 413).
Comment: Figure 11 Relocated letters and numbers above bars, BG or BS ?
Author’s response: The letters have been relocated and BG has been replaced by BS.
Comment: Line 538 - Add year of publication in Ref. 23
Author’s response: Done
Comment: Given the presence of enough references published in the last 10 years, reduce the number of older references by at least 20%
Author’s response: According to the reviewer’s comment, the following references has been removed: Uppal and Bains. J. Food Sci.Technol. 2012; Bau et al. J. Sci. Food Agric. 1997; Mao et al. J. Sci. Food Agric. 2005; Hübner et al. Eur. Food Res. Technol. 2010; Xu et al. J. Sci. Food Agric. 2012, 92, 380-387.
Reviewer 3 Report
The manuscript is interesting but some corrections should be introduced, such as:
Pg. 6, lines 247-248: “Chroma (a21b2)1/2 based in the CIELab (L*, a* and b*). The formula for Chroma should be revised.
Pg. 6, line 253: “…using a scale from 0 to 5.” The meaning of 0 and 5 in the scale should be well described.
Pg. 14, lines 432-433: “increase of flavor in germinated barley. Texture was also modified, increasing the density of the powder with the germination time.” The authors should explain better the sensory analysis results. Example: the increase of flavour means the increase of the intensity of flavour? The density of the powder refers to the perceived density of the powder? The texture evaluation was only associated with the density evaluation? Please improve the explanations.
Author Response
Comment: Pg. 6, lines 247-248: “Chroma (a21b2)1/2 based in the CIELab (L*, a* and b*). The formula for Chroma should be revised.
Author’s response: The formula for Chroma has been revised and corrected, according to the reviewer’s comment (p.6, line 237).
Comment: Pg. 6, line 253: “…using a scale from 0 to 5.” The meaning of 0 and 5 in the scale should be well described
Author’s response: The meaning of 0 (disliked extremely) and 5 (liked extremely), has been including in the revised manuscript (p.6, lines 242-243).
Comment: Pg. 14, lines 432-433: “increase of flavor in germinated barley. Texture was also modified, increasing the density of the powder with the germination time.” The authors should explain better the sensory analysis results. Example: the increase of flavour means the increase of the intensity of flavour? The density of the powder refers to the perceived density of the powder? The texture evaluation was only associated with the density evaluation? Please improve the explanations.
Author’s response: As the reviewer has indicated, the increase of flavor means the increase of the intensity of flavor. Moreover, the discussion of texture evaluation results has been improved, as recommended by the reviewer (p. 14, lines 434-436).
Round 2
Reviewer 1 Report
Manuscript is correct.